# Nasal Septum Changes in Adolescents Treated with Tooth-Borne and Bone-Borne Rapid Maxillary Expansion: A CBCT Retrospective Study Using Skeletal Tortuosity Ratio and Deviation Analysis

**DOI:** 10.3390/children9121853

**Published:** 2022-11-29

**Authors:** Vincenzo Ronsivalle, Elisabetta Carli, Antonino Lo Giudice, Manuel Lagravère, Rosalia Leonardi, Pietro Venezia

**Affiliations:** 1Department of General Surgery and Surgical-Medical Specialties, School of Dentistry, Section of Orthodontics, University of Catania, Via S. Sofia 78, 95124 Catania, Italy; 2Department of Surgical Pathology, Molecular Medicine and Critical Area, University of Pisa, 56126 Pisa, Italy; 3Division of Orthodontics, Faculty of Dentistry, University of Alberta, Edmonton, AB T6G 1C9, Canada

**Keywords:** rapid maxillary expansion, nasal septum deviation, bone-borne RME, tooth-borne RME, orthodontics, skeletal anchorage

## Abstract

Background: Using three-dimensional (3D) images, this study evaluated the impact of Rapid Maxillary Expansion (RME) on changes in Nasal Septal Deviation (NSD). Methods: Cone-beam computed tomography (CBCT) scan of 40 children with transverse maxillary deficiency, who received tooth-borne (TB) RME or bone-borne (BB) RME, were included in this investigation. Two CBCT scans were performed: one before to appliance installation (T0) and one after a 6-month retention period (T1). The analysis was performed by dividing the actual length of the septum by the desired length in the mid-sagittal plane to measure NSD based on the tortuosity ratio (TR). Results: Subjects in the TB group showed a statistically significant reduction (*p* < 0.05) of the TR value from T0 to T1, according to the paired Student t test. Subjects in the BB group showed similar findings, with a statistically significant reduction (*p* < 0.05) of the TR value from T0. No statistically significant differences were found between the mean changes of TR between TB group and BB group. Conclusions: RME may have some effects in reducing the degree of NSD; however, no differences were found between RME performed with TB and BB anchorage systems.

## 1. Introduction

Nasal septal deviation (NSD) is defined as the deviation of either the septal bone or the cartilage or both from the facial midline [1], which can be caused by congenital deformation, traumatic/iatrogenic injury or important nasal infection [2]. NSD can be responsible for nasal obstruction which impairs nasal breathing by reducing nasal airflow and increasing nasal airway resistance [3,4], if chronical, impaired nasal breathing can interfere with craniofacial development in growing subjects, according to Moss’s functional theory [5]. Skeletal openbite, mandibular growth pattern featuring clockwise rotation, with or without mandibular retrognathia, and transverse maxillary deficiency with posterior cross-bite, are often associated with chronical oral breathing [6].

Rapid maxillary expansion (RME) is the main treatment for the correcting transverse maxillary deficiency [7]. Its effects can influence the anatomy and the function of the nasal structures [8], which is of clinical relevance considering that maxillary contraction is one of the most recurrent craniofacial disharmonies necessitating treatment in growing patients [9,10,11]. In particular, previous studies [12,13] showed that RME enlarge the dimension of nasal cavity (about one-third of appliance expansion) and increases its volume by displacing the nasal lateral walls apart. These changes could explain the improvement of the nasal breathing and the reduction of nasal airway resistance often recorded in treated subjects [14,15]. Since RME can influence nasal cavity geometry [16,17], it has been assumed that RME could improve nasal septal deviation during childhood. In this regard, Farronato et al. [17] found NSD reduction in 94% of cases treated with RME; instead, Altug-Atac et al. [18] and Aziz et al. [1] found no differences between pre- and post-treatment conditions. Such conflicting evidence may be attributed to methodological biases due to the inaccuracy of 2D measurements on posteroanterior radiographs or to the inclusion of subjects in adolescence [1,18], which, in turn, could favor dental effects reducing skeletal changes with RME [19].

Miniscrew-assisted maxillary expansion has been proposed to moderate dento-alveolar effects in favor of skeletal changes and, according to recent evidence [20], it would seem that bone-borne maxillary expansion (BB-RME) produce greater skeletal expansion compared to conventional tooth-borne appliance (TB-RME), including effects on nasal cavity dimensions. With this notion in mind, it is important to assess if both protocols may determine different effects on NSD, since there is no evidence on this concern.

Cone-beam computed tomography (CBCT) offers comprehensive anatomical data compared to two-dimensional (2D) images. Concerning nasal anatomy, CBCT scans allow a clearer definition of sinuses and the nasal cavity, and provides a deeper evaluation of the anatomical variants of the deviated nasal septum, concha bullosa, and turbinates [1], allowing a more accurate assessment and comparison of anatomical structures. Accordingly, the present study aimed to investigate the changes in NSD after RME and to compare data findings a between patients treated with tooth-borne and bone-borne maxillary expanders. The null hypotheses were (1) the absence of significant differences in the tortuosity ratio of nasal septum between pre- and post-treatment conditions and (2) the absence of significant differences of the same outcome between the two expansion protocols.

## 2. Materials and Methods

### 2.1. Study Sample

This retrospective study was approved by the Institutional Review Board of Indiana University–Purdue University (IRB protocol number: Pro00075765) and was conducted according to the principles of the Helsinki declaration. All subjects presented bilateral maxillary cross-bite and underwent RME with tooth-borne (TB group) or a bone-borne (BB group) anchorage systems. All subjects signed consent form for the orthodontic treatment. Moreover, the CBCTs used in this study were acquired from previously published materials [21,22], avoiding further radiation exposure to the patients. Inclusion criteria were: (1) age between 8 and 12 years, (2) pre-pubertal stage according to the CVMS method (from stage CS1 to stage CS3) [23], (3) skeletal maxillary transverse deficiency with or without posterior crossbite, 3) availability of pre- and post-retention CBCT scans. Subjects with previous orthodontic treatment, craniofacial, and dental anomalies were excluded as well as CBCT scans of poor quality or featuring artefacts. NSD was discovered as an incidental finding in pre-treatment CBCT scans.

The TB group included 20 subjects (12 females, 8 males) with a mean age of 10.53 ± 0.9 years, while the BB group included 20 subjects (13 females, 7 males) with a mean age of 11.22 ± 1.1 years. The TB group received Hyrax appliance with bands on the first permanent molars and first premolars. In the BB group, the expander was supported by two mini-screws (length: 12 mm; diameter: 1.5 mm) placed between the permanent first molars and second premolars. The expander was activated twice a day (1 turn = 0.25 mm: daily activation = 0.50 mm) in both groups up to the overcorrection of the malocclusion. Finally, the appliance was kept as retention for 6 months. 

### 2.2. CBCT Examinations

Cone Beam Computed Tomography (CBCT) was performed prior to treatment (T0) and after the appliance was removal (T1). Patients were scanned with the same iCAT CBCT unit (Imaging Sciences International, Hartfield, PA, USA). The acquisition protocol included isotropic voxels of 0.3 mm size, 8.9 s, a wide field of view at 120 kV, and 20 mA. The distance between two slices was 0.3 mm. All the image data sets were acquired and saved using the Digital Imaging and Communications in Medicine (DICOM) format on a personal computer workstation for further analysis.

After, DICOM files of all CBCTs were imported into the Dolphin 3D software (Dolphin Imaging1, version 11.0, Chatsworth, CA, USA) to perform the reorientation of the skull according to a validated protocol [24,25] (Figure 1).

The reorientation of the CBCTs taken at T1 required the use of a voxel-based superimposition. The “sub-region box” in Dolphin program was used to pick the anterior cranial base region (Figure 2). The software matched the voxels of each CBCT in this region using thedefined edges and automatically superimposed them between T0 and T1. The CBCT obtained at T1 was then exported as a new DICOM file.

### 2.3. Assessment of Nasal Septum Deviation (NSD)

Nasal septum was traced at two different levels in coronal view, i.e., at the Crista Galli, and (2) at the Anterior Nasal Spine (Figure 3). Tracing was performed in the cranio-caudal direction by placing points 1–2 mm apart [1]. NSD was calculated according to the “degree of tortuosity” or the ratio of length of the curve to the length of an imaginary line in the mid sagittal plane [1], expressed with the formula:(1)$$\mathrm{TR}=\frac{L\;\mathrm{actual}}{L\;\mathrm{ideal}}$$

Both the actual length of the septum and its ideal length were used to calculate TR values [26].

### 2.4. Statistical Analysis

20 subjects (10 in the TB group and 10 in the BB group) were used for a preliminary analysis of sample size power. The results suggested that 16 patients would be needed to achieve the 80% power to identify a mean difference of 0.019 mm in the TR between T0 and T1, with a confidence level of 95% and a beta error level of 20%. However, based on the criteria for inclusion, we were able to enroll 20 participants in each group, which improved the data’s reliability.

The data’s normality was initially tested using the Kolmogorov-Smirnov test. Parametric tests were utilized to assess and compare measures because since the data showed uniform variance. Paired Student’s *t*-test was used to compare the degree of tortuosity of the nasal septum between T0 and T1 data, in both TB and BB groups. The post-treatment changes of the NSD were compared between the two groups using the Student’s *t*-test. The intraclass correlation coefficient (ICC) was used to measure inter-examiner reliability, while Dahlberg’s formula was used to measure technique error. The statistical program SPSS^®^ (version 24 IBM Corporation, 1 New Orchard Road, Armonk, NY, USA) was used to evaluate data sets.

## 3. Results

Pre- and post-treatment data of real length, ideal length, and TR of the nasal septum are reported in Table 1. Subjects in the TB group showed a statistically significant reduction (*p* < 0.05) of the TR value from T0 (mean value = 1.052; SD = 0.032) to T1 (mean value = 1.014; SD = 0.011), according to the unpaired Student t test (Table 2). Subjects in the BB group showed similar findings, with a statistically significant reduction (*p* < 0.05) of the TR value from T0 (mean value = 1.054; SD = 0.030) to T1 (mean value = 1.021; SD = 0.012) (Table 2). These findings would suggest a slight straightening of the nasal septum after RME in both group (Table 2).

No statistically significant differences were found between the mean changes of TR between the TB group (mean value = 0.038; SD = 0.034) and BB group (mean value = 0.033; SD = 0.021) groups, according to the unpaired Student *t*-test.

Concerning the reliability of the methodology, no differences were found between the two readings of TR with an excellent correlation index found for intra-operator readings (0.991) and for inter-operator readings (0.978). According to the Dahlberg’s formula, the random error for the calculation of TR was 0.003 mm.

## 4. Discussion

To the best of our knowledge, this is the first study in the literature that comparatively assesses the effect of RME on NSD between tooth-bone and bone-borne anchorage systems. Furthermore, considering the paucity of studies regarding the effect of RME on nasal septum changes, the present study provides new evidence on this topic.

In the present study, NSD was discovered as an incidental finding in CBCT scans before treatment. We measured the changes of the nasal septum by using a quantitative approach. In particular, we calculated the degree of tortuosity (TR) of the nasal septum at T0 and T1, and we performed intra-group and inter-group comparison to quantify the treatment effect. TR has been described as a valid method for assessing changes of NSD since it solely measures the nasal septum and excludes other confounding nasal pathology that could affect septal deviation, such as turbinate hypertrophy or mucosal swelling [1].

According to the present findings, RME had some limited significant effects on the degree of nasal septum deviation in the medium-term (post retention), either with tooth-borne or bone-borne appliances in place. In both groups, NSD was slightly reduced when comparing pre- and post-treatment values of TR, which would suggest some potential changes in NSD. The small sample size and individual patient variation could have hampered to reach a higher statistical significance of the data. Given that the administration of RME in pre-pubertal stage would favor skeletal changes compared to dento-alveolar effects [27,28], and considering that all the subjects included in the present study were in childhood and pre-pubertal skeletal growth stages, it is likely that the nasal septum changes were the result of a not-advanced craniofacial development [29]. In this regard, the studies that reported favorable effects of RME on NSD had recruited subjects prior to their adolescent growth spurt [17,30] and in mixed dentition, that is, when maxillary expansion can be attributed one half to two thirds rather to skeletal changes [14,31]. Future studies should evaluate the effect of RME on the changes in the degree of NSD according to different growth stages. Moreover, since there is no “gold standard” test to diagnose septal deviation and studies have used different protocols for measuring septal deviation, such methodological inconsistency may have contributed to the contrasting findings reported in the literature [3].

Few studies, including the present, have assessed the effect of RME on NSD, but several scientific contributions have addressed the influence of RME on nasal airway and respiratory performance. Most of the studies have confirmed that RME show promising results of effectiveness on the airway dimension, both in the short-term and in the long-term [18,32,33]. In particular, a significant reduction in nasal resistance is observed with functional examinations which was associated with the increased nasal cavity width and volume after RME [34].

A recent well-conducted study [35] used the Computational Fluid Dynamics test to investigate the changes of nasal airflow in children with NSD and maxillary transverse deficiency. According to their findings, the nasal airflow became more symmetric after RME, and occupied the middle and inferior region of the common meatus, which represents the main region of heat exchange from mucosa [36]. Also, the velocity of the nasal cavity flow was decreased, improving the heating/cooling and humidification processes of inspired air. Although the Computational Fluid Dynamics test is based on 3D virtual representation of the fluid dynamic, the mentioned study [35] provides an indication of the potential functional benefits of RME on respiratory performance in subjects affected by NSD. Our findings would also suggest that RME could reduce the severity of NSD; however, we have no functional data (virtual or clinical) to compare with the skeletal changes detected.

Recent evidence would suggest that bone-borne expanders seem to produce greater orthopedic effects, involving the nasal anatomy, compared to tooth-borne expanders [37]. This discrepancy may be explained by the different treatment approaches used, since tooth-borne expanders are anchored to the dentition while bone-borne expanders have a bone-to-bone contact. This can affect the amount of tensile force transmitted to the surrounding structures. In light of this, it would be helpful for clinicians to know whether bone-borne RME can affect NSD differently from tooth-borne RME because it would guide them in selecting the type of expander appliance in accordance with the specific clinical conditions and indications provided by the otolaryngologist. According to the present findings, the null hypothesis was confirmed since no differences were found between the TB and BB group in the changes of NSD. Thus, clinicians should not expect a greater improvement of NSD with one of the two anchorage protocols used. Considering the small age differences between TB and BB groups in this study, it is possible that skeletal data findings were influenced by a similar maturation stage of the midpalatal suture and similar resistances from circummaxillary sutures. In this regard, future studies are warmly encouraged to test and compare groups of subjects with higher mean age differences, in order to verify possible significant skeletal differences after TB and BB treatment.

Based on the small improvement of NSD recorded in the present investigation, the clinical implications of the present findings remain questionable, especially in the absence of comparative functional respiratory data. Further studies evaluating the NSD changes and the functional respiratory performance after RME are warmly encouraged. Attention should be applied to the consistency of the methodology used involving RME activation, anchorage systems, methods for nasal airway change assessment, skeletal maturation stage individual patient, and concurrent pathologies affecting nasal soft tissues.

### Limitations

The lack of a control group is the main drawback of the study. Changes in the NSD may have been induced also by normal growth in both groups. However, this limitation could be considered negligible since data were obtained after short-term evaluation (6 months). Moreover, the administration of CBCT scans to control subjects would have introduced ethical concerns due to additional radiation exposure to the patients [38,39].It must be underlined that the CBCT protocol included isotropic voxel size of 0.3 mm and we cannot exclude a slight underestimation of the nasal septum. Since patients in both groups were scanned with the same CBCT machine, this limitation did not affect the reliability of comparative data.

## 5. Conclusions

RME would determine a small reduction of the nasal septum tortuosity ratio (TR).The anchorage system, i.e., the usage of tooth-borne (TB) or bone-borne (BB) expanders, did not influence the effect of RME on nasal septum deviation (NSD).RME may have some effects in reducing the degree of NSD; however, no differences were found between RME performed with TB and BB anchorage systems.

## Figures and Tables

**Figure 1 children-09-01853-f001:**
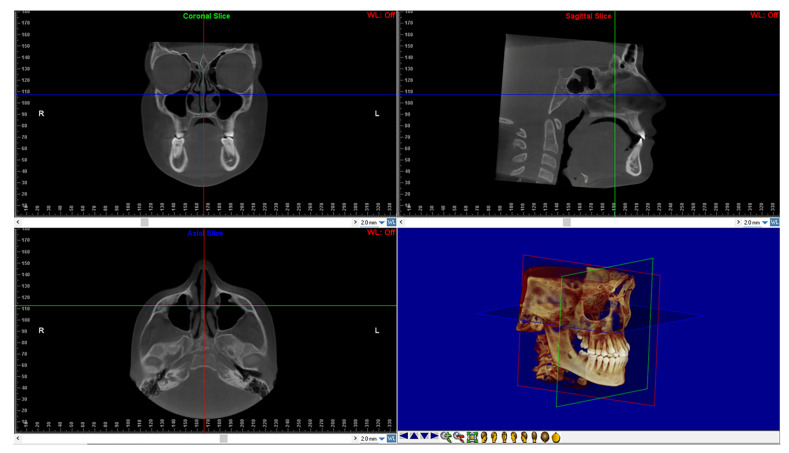
Head reorientation in the axial, sagittal, and coronal planes of CBCT scans. The 3D image shows the head orientation in 3D space.

**Figure 2 children-09-01853-f002:**
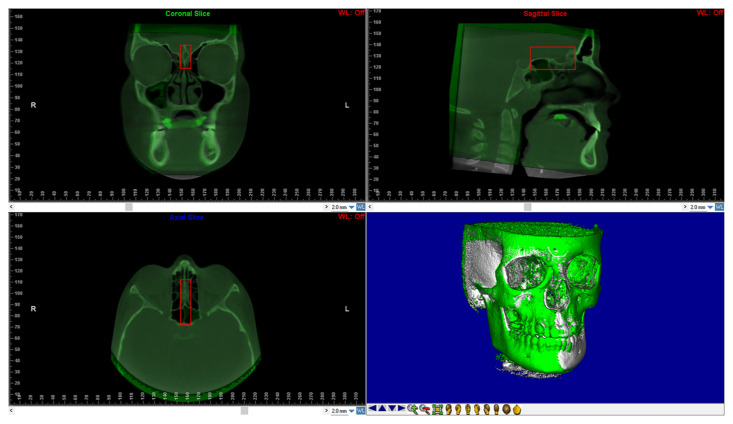
Shows the anterior cranial base using voxel-based superimposition. The area of the cranial base to be used as a stable reference for the superimposition is defined in three dimensions using the red box.

**Figure 3 children-09-01853-f003:**
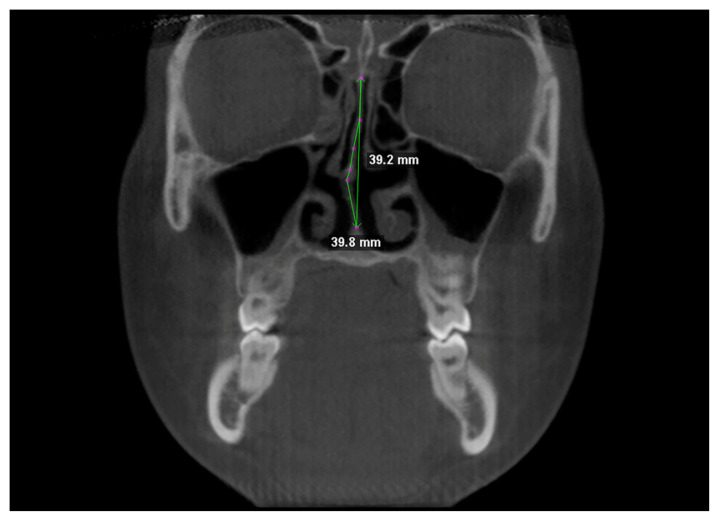
Linear measurements for calculating the tortuosity ratio (TR).

**Table 1 children-09-01853-t001:** Inferential Statistics of the Tortuosity Ratio (TR) of the nasal septum before treatment (T0) and after retention (T1) stages.

	Timing		Mean TR (mm)		95% CI		
		N		SD	Upper Limit	Lower Limit	Significance *
TB	T0	20	1.052	0.032	1.067	1.037	*p* < 0.05
	T1	20	1.014	0.011	1.019	1.009	
BB	T0	20	1.054	0.030	1.068	1.040	*p* < 0.05
	T1	20	1.021	0.012	1.026	1.015	

TB, tooth-borne group; BB indicates bone-borne group; CI, coefficient interval; N, sample number; SD, standard deviation. * Significance set at *p* < 0.05 and based on paired Student’s *t*-test.

**Table 2 children-09-01853-t002:** Comparisons of mean changes of the nasal septum Tortuosity Ratio (TR) between Tooth-Borne (TB) and Bone-Borne (BB) groups.

Groups	N	Mean Differences (mm)	SD	Median Differences (mm)	Minimum	Maximum	Significance *
TB	20	0.038	0.034	0.028	0.001	0.133	NS
BB	20	0.033	0.021	0.037	0.001	0.069	

N indicates sample number; SD, standard deviation. * Significance set at *p* < 0.05 and based unpaired Student *t*-test.

## Data Availability

The datasets used and analyzed during the current study are available from the corresponding author on reasonable request.

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
