# Peer review of "Nasal Septum Changes in Adolescents Treated with Tooth-Borne and Bone-Borne Rapid Maxillary Expansion: A CBCT Retrospective Study Using Skeletal Tortuosity Ratio and Deviation Analysis"

_children, 2022, doi:10.3390/children9121853_

Round 1

Reviewer 1 Report

Dear Authors,

In my opinion, the main finding of the study is that the nasal septum position improves slightly after RME. It is debatable to conclude that the changes after TB at this age are significantly different from the changes after BB, due to the slight age difference between the study groups and the significant probability of skeletal expansion of the maxilla also with TB.

For this reason, please emphasize the fact that in order to draw more reliable conclusions, it is necessary to find groups of subjects with a higher mean age, so that it is possible to notice significant skeletal differences after TB and BB treatment.

Apart from that, the methodology is correct. In the results section, please correct the numbers in the tables by replacing the comma with a dot, for example 1.052 instead of 1,052.

In the current version of the manuscript, both tables have been truncated by the margins and are therefore illegible - please correct this. Moreover, in Table 2, I do not understand the Timing column - why T0 is given for the TB group, and T1 is given for the BB group?

In the limitations and conclusions sections, please remove the hyphens that start the paragraphs. Please prepare your conclusions in the form of a bulleted list.

There are no references added to the manuscript - please include it! Without it, I am not able to check the credibility of the citations.

Author Response

We thank the reviewer for his/her suggestions that have improved the quality of our manuscript.

Reviewer’s concern 1. In my opinion, the main finding of the study is that the nasal septum position improves slightly after RME. It is debatable to conclude that the changes after TB at this age are significantly different from the changes after BB, due to the slight age difference between the study groups and the significant probability of skeletal expansion of the maxilla also with TB.

For this reason, please emphasize the fact that in order to draw more reliable conclusions, it is necessary to find groups of subjects with a higher mean age, so that it is possible to notice significant skeletal differences after TB and BB treatment.

Authors’ response. We thank the reviewer for his suggestion. We TOTALLY agree with him/her. In fact, we have already explained at the end of the discussion that the differences between TB and BB were not relevant. However, we should have better clarified the topic addressed by the reviewer, that is, data findings could have been influenced by similar age and maturation stage.  In this regard, we have reported in the text: “Considering the small age differences between TB and BB groups in this study, it is possible that skeletal data findings were influenced by similar maturation stage of the midpalatal suture and similar resistances from circummaxillary sutures. In this regard, future studies are warmly encouraged to test and compare groups of subjects with a higher mean age differences, in order to verify possible significant skeletal differences after TB and BB treatment.”

Again, we warmly thank the reviewer for his suggestion.

Reviewer’s concern 2. Apart from that, the methodology is correct. In the results section, please correct the numbers in the tables by replacing the comma with a dot, for example 1.052 instead of 1,052.

Authors’ response. We thank the reviewer for his suggestion. According to the reviewer’s request, we’ve made appropriate changes.

Reviewer’s concern 3. In the current version of the manuscript, both tables have been truncated by the margins and are therefore illegible - please correct this. Moreover, in Table 2, I do not understand the Timing column - why T0 is given for the TB group, and T1 is given for the BB group?

Authors’ response. According to the reviewer’s request, we’ve made appropriate changes. Thanks for this suggestion since we had accidentally inserted the timing column.

Reviewer’s concern 4. In the limitations and conclusions sections, please remove the hyphens that start the paragraphs. Please prepare your conclusions in the form of a bulleted list.

Authors’ response. According to the reviewer’s request, we’ve made appropriate changes.

Reviewer’s concern 5. There are no references added to the manuscript - please include it! Without it, I am not able to check the credibility of the citations.

Authors’ response. We thank the reviewer for his/her suggestions. We have noticed that the .pdf version of the manuscript lacked of reference, probably something went wrong in the generation of the .pdf file. We apologize for the mistake.

Reviewer 2 Report

Dear authors,

thank you for submitting your paper.

The study is well conducted but before publication some changes are required.

-In the Abstract please remove "will evaluate" and use " evaluated" as this is a retrospected study.

-Reference were not available in the manuscript 

-Please when you report the effectiveness on the airway dimension  consider tu add the following reference:

https://doi.org/10.3390/ma13102239

- you stated " The absence of a control group is the main limitation of the present study." 

is this referred to an untreated control group?  as the present study compared two
groups of children treated with two different modalities.

Author Response

We thank the reviewer for his/her suggestions that have improved the quality of our manuscript.

Reviewer’s concern 1. In the Abstract please remove "will evaluate" and use " evaluated" as this is a retrospected study.

 Authors’ response. We thank the reviewer for his/her suggestions. According to the reviewer’s request, we’ve made appropriate changes.

 Reviewer’s concern 2. Reference were not available in the manuscript.

 Authors’ response. We thank the reviewer for his/her suggestions. We have noticed that the .pdf version of the manuscript lacked of reference, probably something went wrong in the generation of the .pdf file. We apologize for the mistake.

 Reviewer’s concern 3. Please when you report the effectiveness on the airway dimension consider tu add the following reference: https://doi.org/10.3390/ma13102239.

 Authors’ response. We thank the reviewer for his/her suggestions. According to the reviewer’s request, we’ve made appropriate changes. 

Reviewer’s concern 4. You stated " The absence of a control group is the main limitation of the present study." Is this referred to an untreated control group?  as the present study compared two groups of children treated with two different modalities.

 Authors’ response. We thank the reviewer for having addressed this concern. Yes, we refer to the absence on untreated group of subjects that would have provided potential information about the skeletal nasal changes due to growth.

Reviewer 3 Report

Thank you for the opportunity to review this paper.

The paper is well structured and well-written with a clear and interesting research question which was executed well with sound scientific methodology

I have no further comments

Wishing the authors all the best!

Author Response

We thank the reviewer for his/her suggestions that have improved the quality of our manuscript.

 Reviewer’s general consideration. Thank you for the opportunity to review this paper. The paper is well structured and well-written with a clear and interesting research question which was executed well with sound scientific methodology I have no further comments Wishing the authors all the best!

Authors’ response. We thank the reviewer for his/her positive response and for having underlined the efforts in performing this study. Thanks again.

Round 2

Reviewer 1 Report

Dear Authors,

thank you for correcting the manuscript according to my comments. Now the article, in my opinion, can be published. Please also check the stylistic aspect - for my part, I suggest replacing the commas with dots in the decimal notation within the entire manuscript (not only in the tables, which was corrected at my request). I wish you all the best!

Author Response

We warmly thank the reviewer for the suggestion. Accordingly, we've consistently replaced commas with dots.

thanks you again